# Evolutionary Rates, Divergence Rates, and Performance of Individual Mitochondrial Genes Based on Phylogenetic Analysis of Copepoda

**DOI:** 10.3390/genes14071496

**Published:** 2023-07-22

**Authors:** Junzong He, Zhihao Zhou, Yan Huang, Jinmei Feng, Wenxiang Li, Guitang Wang, Congjie Hua

**Affiliations:** 1School of Life Science, Jianghan University, Wuhan 430056, China; 202212101136@stu.jhun.edu.cn (J.H.); 2816603455@stu.jhun.edu.cn (Z.Z.); 202212101113@stu.jhun.edu.cn (Y.H.); 2Department of Pathogenic Biology, School of Medicine, Jianghan University, Wuhan 430056, China; fengjm@jhun.edu.cn; 3Institute of Hydrobiology, Chinese Academy of Sciences, Wuhan 430056, China; liwx@ihb.ac.cn (W.L.); gtwang@ihb.ac.cn (G.W.); 4Wuhan Institutes of Biomedical Sciences, School of Medicine, Jianghan University, Wuhan 430056, China

**Keywords:** mitochondrial genome, phylogenetics, Copepoda, molecular marker, *Sinergasilus*

## Abstract

Copepoda is a large and diverse group of crustaceans, which is widely distributed worldwide. It encompasses roughly 9 orders, whose phylogeny remains unresolved. We sequenced the complete mitochondrial genome (mitogenome) of *Sinergasilus major* (Markevich, 1940) and used it to explore the phylogeny and mitogenomic evolution of Copepoda. The mitogenome of *S. major* (14,588 bp) encodes the standard 37 genes as well as a putative control region, and molecular features are highly conserved compared to other Copepoda mitogenomes. Comparative analyses indicated that the *nad2* gene has relatively high nucleotide diversity and evolutionary rate, as well as the largest amount of phylogenetic information. These results indicate that *nad2* may be a better marker to investigate phylogenetic relationships among closely related species in Copepoda than the commonly used *cox1* gene. The sister-group relationship of Siphonostomatoida and Cyclopoida was recovered with strong support in our study. The only topological ambiguity was found within Cyclopoida, which might be caused by the rapid evolution and sparse taxon sampling of this lineage. More taxa and genes should be used to reconstruct the Copepoda phylogeny in the future.

## 1. Introduction

Complete mitochondrial genomes (mitogenomes) have been widely used to investigate molecular evolution and phylogenetic relationships among different lineages of Metazoa due to their haploid nature, a lack of recombination, maternal inheritance, and rapid evolutionary rate compared to their nuclear DNA [1,2]. Metazoan mitogenomes are typically double-stranded, circular molecules that encode 13 protein-coding genes (PCGs), 22 transfer RNA (tRNA) genes, and two ribosomal RNA (rRNA) genes, plus a control region [1,2,3,4]. It is generally believed that using the mitogenome is advantageous for phylogenetic studies in Metazoan [5]. Previous studies have shown that phylogenomic analyses with larger numbers of genes, up to all 37 mitochondrial genes, produce better-supported nodes in comparison to phylogenetic analyses based on a single locus or a few loci [6,7,8]. As technology continues to advance, there is a growing acceptance and adoption of next-generation sequencing techniques. Consequently, obtaining and analyzing complete mitochondrial genome sequences has become increasingly accessible and convenient. Despite the availability of complete mitochondrial genome sequences, many researchers still tend to selectively sequence specific mitochondrial genes and combine them with nuclear loci for conducting phylogenetic analyses. However, the reliability of phylogenetic studies depends in a large part on the appropriate choice of markers [9]. Thus, deciding which genes to use to infer a mitochondrial gene tree remains an important issue in phylogenetics. To achieve this, it is necessary to evaluate the performance of individual mitochondrial genes in a phylogenetic context for specific groups and clades, but no systematic survey has yet addressed this problem in the class Copepoda (Crustacea).

Copepods, which encompass an extensive collection of aquatic crustaceans, are the most abundant metazoans within aquatic ecosystems. They inhabit nearly all freshwater environments and play a significant role as a major constituent of various planktonic, benthic, and groundwater communities [10,11]. Copepods of the genus *Sinergasilus* are typically ectoparasites found on fish. They were later distributed worldwide along with their hosts. There are several “local” genera from the same order worldwide [12,13]. *Sinergasilus major* (Markevich, 1940) is commonly found on the gills of *Ctenopharyngodon idella* (Valenciennes, 1844). As its host has rapidly spread throughout aquatic ecosystems worldwide, this parasite has followed suit, leading to its widespread presence [14]. The highest level of infection is recorded throughout the summer period, with the maximum level of prevalence in September [15]. *Sinergasilus* parasites cause gill tissue swelling and necrosis. Since gills are the respiratory organ of fish, these changes affect their physiological state. For example, in circumstances of oxygen impoverishment in water, fish are more likely to suffer asphyxia due to the reduced area of functional gill tissue. This may negatively affect the aquaculture industry [16,17]. Moreover, due to the extreme diversity in body forms and limited genetic information available for Copepoda, the phylogenetic relationships and taxonomic status of Copepoda remain controversial [18,19,20,21]. This hampers studies of this clade as well-resolved phylogenies are essential to interpretation in biological research [22].

The number of sequenced mitogenomes of Copepoda has greatly increased in recent years, but most previously published molecular phylogenetic studies were based on single loci (mostly *18S*, *28S*, and *cox1* genes) [18,23,24,25]. There is evidence that, due to the low level of polymorphism, the mitochondrial *cox1* gene is not suitable for inferring phylogeny in many metazoan lineages, including copepods [26,27]. The utilization of complete or nearly complete mitogenomes has shown significant advancements in increasing the effectiveness of phylogenetic analyses and the precision of taxonomic relationships, especially when compared to single gene loci [28,29]. To further explore this, the complete mitogenome of *S. major* was sequenced and analyzed, followed by the incorporation of 18 additional complete mitogenomes of Copepoda to reconstruct the phylogeny of the entire clade (Table 1). In addition, the relative phylogenetic performance of each mitochondrial gene and correlated characteristics (nucleotide diversity, Ka/Ks ratio, and gene length) were also investigated. This provided valuable data resources for a more comprehensive comparative analysis of the genomic structure, base composition, substitution, and evolutionary rates in Copepoda. Moreover, by comparing the phylogenetic performance of different genes, the potential and resolution of these molecular markers were demonstrated, which allowed us to identify genes with the best phylogenetic resolution and provide directions for future mitochondrial phylogenetic research in Copepoda.

## 2. Materials and Methods

### 2.1. Sampling and Identification

Specimens of *S. major* were collected from *C. idella* at a farm in Changsha, Hunan, China (28°17′ N, 113°04′ E), and preserved in 90% ethanol at −20 °C until DNA extraction. All samples were identified initially using traditional morphological keys [30]. In order to obtain genomic DNA, an adult specimen was used, and the extraction process was carried out using an animal tissue genomic DNA extraction kit from TIANGEN Biotech, Beijing, China, following the manufacturer’s instructions. The quality of the extracted DNA was assessed by visualizing it on 1% agarose gels. A partial sequence of the 18S sequence (~1300 bp) for species identification was amplified with the primers 18SF: (5′-AAGGTGTGMCCTATCAACT-3′) and 18SR: (5′-TTACTTCCTCTAAACGCTC-3′), and sequenced by Sangon Biotech (Co., Ltd., Shanghai, China) using the Sanger sequencing methodology. The NCBI’s BLAST program was used for the homology search, and the sequences deposited within the NCBI database were browsed. The GenBank accession number for the 18S gene fragment of the isolated DNA sample used in this work for mitochondrial DNA sequencing is OP076956.

### 2.2. Mitochondrial Genome Sequencing and Assembly

The mitogenome of *S. major* was sequenced by Sangon Biotech (Shanghai, China) Co., Ltd., using Sanger sequencing; the nine pairs of species-specific primers are provided in Appendix A. The amplicons were assembled using Geneious [31]. The MITOS2 web server [32] (available at http://mitos2.bioinf.uni-leipzig.de/index.py; accessed on 1 May 2022) was used for the annotation of the newly sequenced mitogenome. PCGs were identified by analyzing the open reading frames of the genomic DNA using the invertebrate mitochondrial genetic code. Additionally, alignment with the orthologous PCG sequences of Copepoda available in the GenBank database was performed to confirm their identity and ensure accuracy (available at https://www.ncbi.nlm.nih.gov/genbank/; accessed on 12 May 2022) (Table 1). The tRNA genes were annotated using the MITOS2 web server, and their secondary structures were further predicted by using the programs TRNAscan-se (http://lowelab.ucsc.edu/tRNAscan-SE/; accessed on 13 May 2022) and ARWEN (http://130.235.244.92/ARWEN/; accessed on 15 May 2022) [33]. Finally, all genes were manually verified after the annotation, and all gene overlaps were carefully examined for putative annotation artefacts.

### 2.3. Genome Composition and Sequence Analyses

The CGView Server was used to generate a circular map [34]. The PhyloSuite program [35] was employed to analyze and extract the recorded annotations from the Word documents. Additionally, this program was used to facilitate the generation of GenBank submission files and organization tables for the mitogenomes. A similar software tool was utilized to conduct comparative genomics analysis on the mitogenomes of Copepoda, including examination of codon usage, amino acid proportions, and relative synonymous codon usage (RSCU). The base skew heatmap was drawn using the ggplot2 package [36] implemented in the R program, with the assistance of the statistics file generated by the PhyloSuite program. Eighteen complete mitogenomes (Table 1) of Copepoda, including one species sequenced for this study, were used to calculate nucleotide diversity (Pi), non-synonymous substitutions (Ka), and synonymous substitutions (Ks) using the DnaSP program version 6 [37]. The nucleotide diversity (Pi) analysis of PCG and rRNA genes was conducted using a sliding window of 200 bp with a step size of 20 bp.

### 2.4. Phylogenetic Analyses

Aside from the *S. major* mitogenome sequenced in this study, 7 partial (the total size larger than 10,000 bp) and 17 complete Copepoda mitogenomes were procured from GenBank for the phylogenetic analyses. Nine outgroup species were also included: three species each of Branchiopoda, Malacostraca, and Thecostraca. The presence of incomplete sequences might cause erroneous estimates of topology, node support, and branch lengths [38], so two datasets to test the impact of incomplete sequences were designed: subgroup_1 (comprising incomplete mitogenomes) and subgroup_2 (only complete mitogenomes). This also allowed us to assess the impact of sparse taxon sampling on phylogenetic reconstruction [39]. The GenBank accession numbers are listed in Table 1.

For both subgroups, nucleotide sequences were individually aligned using the Normal mode in MAFFT v7.0 [40]. All alignments were checked manually, and ambiguously aligned regions were removed using the Gblocks program [41]. Subsequently, 37 genes (13 PCGs + 2 rRNA genes + 22 tRNA genes) were concatenated in the PhyloSuite program [35]. Two concatenated matrixes were used to construct the phylogenetic trees using the algorithms incorporated into two PhyloSuite plug-in programs: maximum likelihood (ML) in IQ-tree v.1.6 [42], and Bayesian inference (BI) in MrBayes program, version v3.2.6 [43]. The best-fit substitution models and partitioning schemes were inferred using the ModelFinder program [44] and PartitionFinder 2 program [45], respectively. The Bayesian information criterion (BIC) was utilized to select the best models by employing a “greedy” search with linked branch lengths. The best schemes, models, and other parameters are provided in Appendix A. For the IQ-TREE ML analyses, 10,000 ultrafast bootstrap replicates were conducted using the substitutional models selected with the “mtZOA” option. The BI analyses consisted of 2 million generations with 4 chains, sampling every 315 generations, and a burn-in of 25% of the sampled values. Stationarity was considered to be reached when the average standard deviation of the split frequencies fell below 0.01. The phylogenetic trees were visualized and edited using the iTOL web server [46] (https://itol.embl.de; accessed on 18 May 2022).

### 2.5. Phylogenetic Examination of Separate Genes

To assess the individual contribution of each mitochondrial gene in constructing the phylogenetic tree, the Ktreedist program [47] was employed. This program was used to evaluate the relative importance of each mitochondrial gene in the overall phylogenetic analysis. The ML analyses were carried out using individual genes and concatenated genes. Because the mitogenome of *Paracyclopina nana* (Smirnov, 1935) lacked the *atp8* gene, this gene was not used in the analyses. The k-scores were calculated via a comparison with a reference tree, i.e., the tree based on the ML analysis of the partitioned dataset comprising all 37 genes of subgroup_2.

## 3. Results

### 3.1. The Structure of the Mitochondrial Genome

The complete mitogenome of *S. major* is a circular, double-stranded DNA molecule of 14,588 bp in length (Figure 1). The sequence is deposited in GenBank under the accession number OQ160840. It contains the typical set of 13 PCGs, 2 rRNA genes, and 22 tRNA genes, as well as a putative control region (CR; Figure 1). A total of 11 genes were transcribed from the majority H-strand, which included four PCGs (*nad4*, *cox1*, *cytb*, and *nad5*) and six tRNA genes (*trnM*, *trnP*, *trnD*, *trnE*, *trnS2*, and *trnY*), whereas the remaining 26 genes, comprising nine PCGs, sixteen tRNA genes, and two rRNA genes (*rrnL* and *rrnS*), were located on the L-strand (Appendix A). The overall nucleotide composition was 33.5% A, 35.6% T, 14.1% G, and 14.7% C, which reveals a strong AT bias (71.1%) (Appendix A). The nucleotide skew statistics showed negative AT-skew (−0.002) and negative GC-skew (−0.021).

The mitogenome of *S. major* had a relatively compact structure, with relatively short intergenic sequences (0 to 29 bp), except for the region between *trnT* and *trnH* (180 bp) and the putative CR (Appendix A). All 13 PCGs were initiated by the typical ATN start codons (nine with ATA, three with ATG, and a single gene with ATC). Most PCGs terminated with TAA or TAG, whereas *cox3* and *cox1* used an incomplete stop codon T (Appendix A). Incomplete stop codons are frequently observed in Copepoda mitogenomes and can be corrected through post-transcriptional polyadenylation [48]. The relative synonymous codon usage (RSCU) values of the protein-coding genes (PCGs) in *S. major* are presented in Appendix A. The PCGs collectively encode a total of 3360 amino acids. The analysis of the codon usage pattern of these PCGs showed that codons encoding leucine (16.41%), serine (10.88%), proline (9.67%), and phenylalanine (8.8%) were the most frequently used, while codon encoding cysteine was rare (0.88%).

The tRNA genes of the newly sequenced species ranged from 54 bp to 74 bp in size. The secondary structure prediction indicated that most tRNA genes could be folded into the typical cloverleaf structure, except for seven tRNA genes. In *trnA*, *trnD*, *trnG*, and *trnV*, the TψC (T) arm was replaced with a simple loop due to unmatched base pairs; moreover, *trnR*, *trnS1*, and *trnS2* completely lacked the dihydrouridine (DHU) arm, whereas *trnC* lacked the TψC (T) arm. The predicted secondary structures for the twenty-two tRNA genes of *S. major* are shown in Appendix A. Additionally, except for the normal Watson–Crick base pairs (A-T and G-C) and G-U matches, a total of seven mismatched base pairs were found, including four A-C base pairs in *trnW* and *trnV*, two A-C base pairs in *trnT*, two U-U base pairs in *trnP* and *trnY*, and one U-C base pair in *trnT*. Such mismatches are probably corrected through post-transcriptional RNA editing [49,50]. Two rRNA genes (*rrnL* and *rrnS*) were transcribed from the L-strand and exhibited a positive GC-skew of 0.083 (Appendix A).

### 3.2. Comparative Analysis of Copepoda Mitogenomes

The mitogenomes of *S. major* and the other 25 Copepoda (18 complete mitogenomes and 7 partial mitogenomes) were compared. The mitogenomes ranged in size from 13,440 bp (*Caligus clemensi* Parker & Margolis, 1964) to 28,462 bp (*Lamproglena orientalis* Markevich, 1936). This is due to *L. orientalis* possessing a duplicated mitogenome. Besides that, length diversification was detected in the rapidly evolving non-coding regions in different species, ranging in size from 38 bp (*Amphiascoides atopus *Lotufo & Fleeger, 1995) to 13,528 bp (*Calanus simillimus* Giesbrecht, 1902). This is the primary contributor to the variations in mitogenome sizes in Copepoda.

The nucleotide composition of the 25 Copepoda mitogenomes showed A + T bias, ranging from 53.7% in *Tigriopus kingsejongensis* to 75.5% in *C. clemensi*. The genus of *Sinergasilus* had a comparatively high A + T content among the 25 selected Copepods: 71.4% in *S. polycolpus*, 71.1% in *S. major*, and 70.4% in *S. undulatus* (Appendix A). The GC-skew values of 13 PCGs were relatively diverse in the 25 Copepoda mitogenomes, ranging from −0.778 to 0.58 (Figure 2). Conversely, there was no significant difference in AT-skew values among the 13 PCGs.

In Copepoda, there are eight types of start codons, and we found that *atp6* and *cox3* have a relative “preference” for the ATG start codon (Figure 3). GTT and CTA are atypical start codons identified in *Labidocera rotunda* (Mori, 1929) (*cox2*) and *Schizopera knabeni* (Lang, 1965) (*nad5*), respectively. All Copepoda shared the same three stop codons: TAA (63.88%), TAG (18.73%), and incomplete codons TA (2.01%) or T (15.38%) (Figure 3).

### 3.3. Nucleotide Diversity and Evolutionary Rate Analysis

The plot of sequence variation ratio exhibits variable nucleotide diversity, with Pi values for the 200 bp windows ranging from 0.273 to 0.465 (Figure 4a). The Ka/Ks ratio for all PCGs was <1. The values of Ka/Ks were sequentially *nad4L* > *atp8* > *nad6* > *nad2* > *nad3* > *nad5* > *nad4* > *atp6* > *nad1* > *cox2* > *cytb* > *cox3* > *cox1* (Figure 4b).

### 3.4. Phylogenetic Analyses

The ML and BI phylogenies reconstructed using the subgroup_1 dataset had similar topologies, with the exception of unstable relationships within the order Cyclopoida (Figure 5). This phylogenetic reconstruction confirmed the monophyly of the class Copepoda. The orders were all monophyletic, with Calanoida as the sister lineage to all other orders: (Calanoida + (Harpacticoida + (Cyclopoida + Siphonostomatoida))). The statistical support for the nodes was high.

Significantly, most previous mitogenomic studies with sparser taxon sampling resolved Harpacticoida as the closest relative of Siphonostomatoida [51,52,53]. A congruent result was obtained in the subgroup_2 dataset analyses (Appendix A).

### 3.5. Phylogenetic Examination of Individual Genes

To better understand the contribution of each mitochondrial gene to the mitochondrial phylogenetic reconstruction, a set of k-scores were calculated using the Ktreedist program (Table 2). The k-scale factor is the ratio between the global divergence (similar to the average branch length) of the individual gene tree and the mitogenomic tree. High scores indicate a poor match between the test and reference trees. The range of k-score is from 0.642 to 3.717; the *nad2* gene obtained the smallest numerical value, and the rRNA and tRNA genes obtained relatively higher k-scores than all PCGs genes.

## 4. Discussion

In this study, we sequenced and analyzed the complete mitogenome of *S. major* and evaluated the nucleotide diversity, evolutionary rate, and relative contribution of each gene for the phylogenetic reconstruction in Copepoda. The mitochondrial genome of *S. major* has a typical structure, including 13 PCGs, 22 tRNA genes, and two rRNA genes, with a strong AT bias (71.1%). The genome-wide bias toward AT is a common feature of Copepoda mitogenomes [26,54]. Negative AT-skew and GC-skew were found. This is unusual in Copepoda and is indicative of inversions in the strand replication order or, otherwise, disrupted replication mechanism (e.g., multiple origins of replication) in the lineages with the reduced/inverted skews [55].

In light of the results comparing different mitogenomes in Copepoda, we found that all AT-skew values were negative among the 13 PCGs, except for *Vulcanolepas fijiensis* (Chan, Ju & Kim, 2019) (*atp8*) and *Metaplax longipes* (Chan, Ju & Kim, 2019) (*atp8* and *cox2*), with AT-skews of 0.052, 0.032, and 0.021, respectively. It is noteworthy that the second position of the PCGs exhibited the best negative AT skewness compared to different mitogenomic parts (Figure 2). This is often, in all probability, a reflection of the actual fact that codons for hydrophobic amino acid residues, which are functionally most popular for the conformational stability of mitochondrial proteins, principally have T within the second codon position [56].

In the nucleotide diversity analysis, the genes with a comparatively high sequence variability were *nad4L* (0.548) and *nad5* (0.546). By contrast, *cox1* (0.299), *cytb* (0.369), and *cox3* (0.386) had a comparatively low sequence variability. Congruent results were observed based on the Ka/Ks analysis. The smallest Ka/Ks ratio was exhibited by the *cox1* gene (average = 0.097), and the highest was exhibited by the *nad4L* gene (average = 0.564). Therefore, these two genes evolved under a comparatively strong and a relaxed purifying selection process, respectively. Both analyses (nucleotide diversity survey and Ka/Ks ratio) consistently showed that *cox1* was the slowest-evolving and least variable gene. This is the most commonly utilized mitochondrial gene in studies of Copepoda [23,57,58]. The faster-evolving and more variable *nad4L*, *nad2*, and *nad6* genes may be more effective markers to investigate genus-level relationships or closely related species in Copepoda, as rapidly evolving genes are capable of providing a higher resolution for phylogenetically closely related taxa [59,60]. Due to its longer length, the *nad2* gene would be advantageous compared to *nad4L* and *nad6*.

The topologies of the phylogenetic tree reconstructed using the subgroup_1 dataset were identical, with a high support at the order levels (Calanoida + (Harpacticoida + (Cyclopoida + Siphonostomatoida))). However, this relationship partially contradicted previously proposed morphology-based [61,62] and molecular data-based copepod relationships [51,52,53]. Mikhailov et al. (2021) [63] employed four concatenated genes (*18S*, *28S*, *cox1*, and histone H3) sampled from 203 copepod species to reconstruct their phylogenetic relationships. The results suggested that the order-level topology was (Calanoida + (Siphonostomatoida + (Cyclopoida + Harpacticoida))), but Siphonostomatoida was paraphyletic, and the node support values were low. Additionally, this topology of Copepoda was partially rejected by a subsequent molecular study [64], which used a larger number of nuclear protein-coding genes, and obtained the following order-level topology: (Calanoida + (Harpacticoida + (Cyclopoida + Siphonostomatoida))).

The subgroup_2 dataset of sparser taxon sampling restored Harpacticoida as the closest relative of Siphonostomatoida like the previous study, which further indicates that sparse taxon sampling might be responsible for those earlier results [39]. Except for the unstable relationship within the Cyclopoida order in the subgroup_2 dataset, the relationships among the remaining three orders of Copepoda (Calanoida, Harpacticoida, and Siphonostomatoida) were identical to the topology obtained using the subgroup_1 dataset. Although taxon sampling had an impact on the relationships among certain families and genera, it did not affect the monophyly of Copepoda. Our research findings provide robust evidence supporting the monophyly of Copepoda, which is also supported by morphological studies [62,65] as well as large-scale phylogenomic analyses [66,67].

In our study, three distinct topologies were observed within the order Cyclopoida (Appendix A), and all of them had a high or moderate node support. In previous studies of Cyclopoida, Lernaeidae was generally more closely related to Cyclopettidae than to Ergasilidae [20,52,68]. The same results were also observed in our analyses, which might be caused by their rapid evolution or lack of adequate representative mitogenomes for each family. We still have reservations about their position, so more studies with better data are needed to resolve this problem in the future. Taxon sampling plays an important role in the phylogeny reconstruction of Copepoda, and more stable family-level and order-level relationships can be inferred by increasing the lineage coverage. The information presented in this paper will serve as a valuable resource to improve our understanding of copepod evolution and their wide range of ecological adaptations in the future. It also provides new insights into conservation strategies. Considering the sensitivity of tree reconstruction based on mitogenomes to taxon sampling, it is important to further validate the results by incorporating data from nuclear genes. Additional information obtained from nuclear genes will provide a more comprehensive understanding of the evolutionary relationships within Copepoda and help confirm the findings obtained from the analysis of mitogenomic data.

Although the sequencing of complete or nearly complete mitogenomes is neither difficult nor expensive in the era of advanced next-generation sequencing platforms and bioinformatics tools, many researchers typically sequence preferred mitochondrial genes and combine them with nuclear gene data. The choice of the ideal mitochondrial gene is a significant problem in phylogenetic studies. From a phylogenetic perspective, not all mitochondrial genes are equally representative of the complete set; thus, phylogenetic resolution varies among different mitochondrial genes. In the phylogenetic examination of individual genes, we found that *rrnS* and *rrnL* provided a weaker phylogenetic resolution than the tRNAs and PCGs. Among the PCGs, *nad1*, *nad2*, and *cox1* demonstrated a good match with the topology of the reference tree, indicating their reliability in phylogenetic reconstruction. On the other hand, the *nad4L* and *nad3* genes exhibited the least congruent topologies. This evidence suggests that *rrnS* and *rrnL* might contribute less to the reconstruction of Copepoda phylogeny. In contrast, *nad1*, *nad2*, and *cox1* outperformed other mitochondrial genes in terms of accurately reflecting evolutionary relationships. Selecting genes with a strong phylogenetic signal is crucial for accurately reconstructing ancient divergences in Copepoda [69].

As mentioned above, relatively high nucleotide diversity and evolutionary rate were identified in the *nad2* gene, which also has the highest phylogenetic contribution (K-scale = 0.624) and provides better phylogenetic resolution and precision in Copepoda relative to the traditional markers such as the *cox1* gene. In the future, *nad2* may be a better choice as a molecular marker in mitochondrial phylogenetic studies examining Copepoda.

## Figures and Tables

**Figure 1 genes-14-01496-f001:**
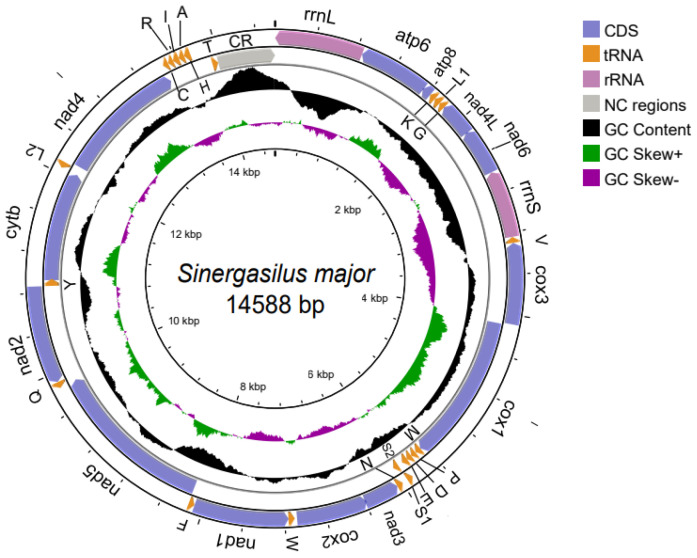
Circular map of the mitogenome for *Sinergasilus major* (Markevich, 1940). Genes or gene regions are highlighted by different colors. tRNAs are represented by single-letter amino acid codes. The outermost circle displays the orientation of gene transcription using arrows. The second circle represents the GC content, with black shading above and below indicating GC content values greater than and less than the genome average, respectively. The third circle represents the GC skew, with green above and purple below denoting GC skew values greater or less than the genome average, respectively. The innermost circle, along with a scale, shows the nucleotide position on the genome.

**Figure 2 genes-14-01496-f002:**
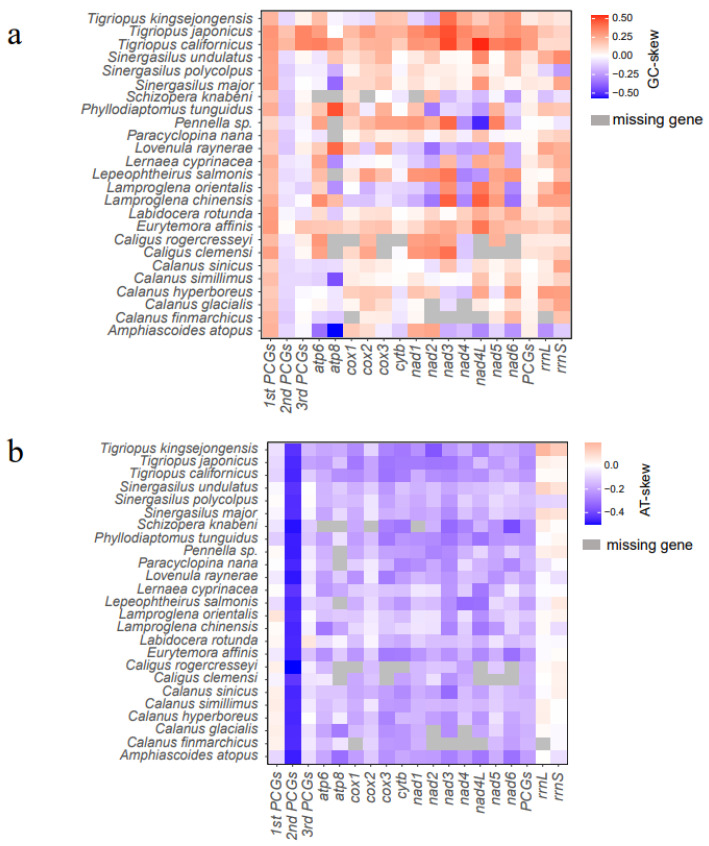
The base skews of various mitogenomic elements among the Copepoda mitogenomes. (**a**) The GC-skew; (**b**) The AT-skew.

**Figure 3 genes-14-01496-f003:**
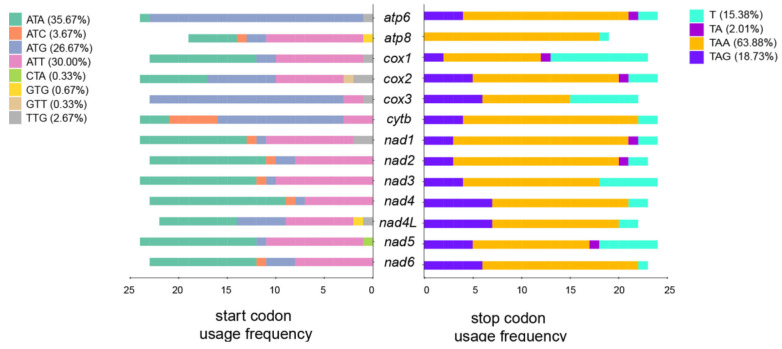
Start codon and stop codon usage for the mitogenomic protein-coding genes of 25 Copepoda species.

**Figure 4 genes-14-01496-f004:**
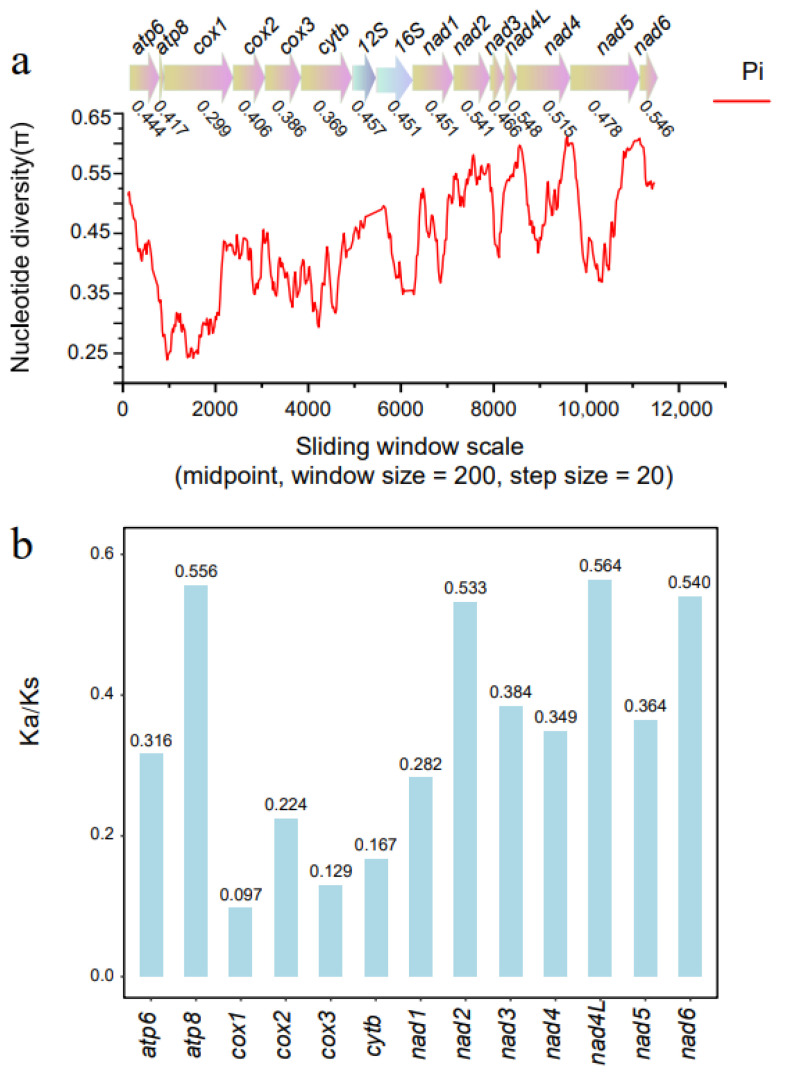
Sliding window analyses of the alignment among 18 complete Copepoda mitogenomes (**a**). The line shows the value of nucleotide diversity (π) in a sliding window analysis with a window size of 200 bp and a step size of 20; the value is inserted at its mid-point. Non-synonymous/synonymous substitution rates (Ka/Ks) of 13 PCGs among the 18 complete Copepoda mitogenomes (**b**).

**Figure 5 genes-14-01496-f005:**
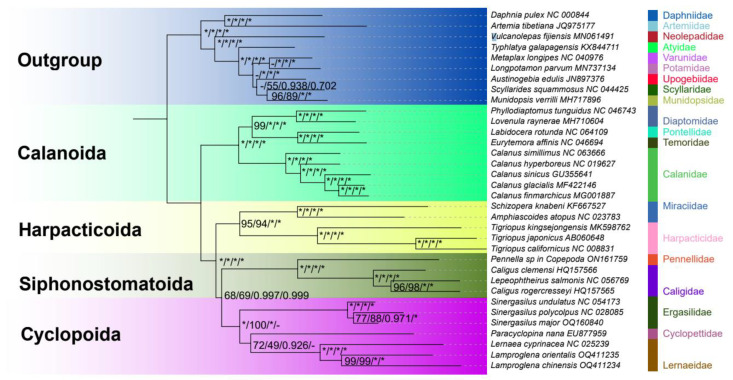
Maximum likelihood (ML) and Bayesian inference (BI) phylogenetic trees inferred from concatenating 13 mitochondrial protein-coding genes, 22 tRNA genes, and two mitochondrial rRNA genes based on subgroup_1 dataset. Values above the nodes represent unpartitioned nucleotide sequences of bootstrap values, partitioned nucleotide sequences of bootstrap values, unpartitioned nucleotide sequences of Bayesian posterior probabilities, and partitioned nucleotide sequences of Bayesian posterior probabilities. “-” indicates not support, and “*” indicates posterior probabilities = 1.00 or ML bootstrap = 100.

**Table 1 genes-14-01496-t001:** Taxonomic information and GenBank accession numbers (IDs) of the mitochondrial genomes used in this study.

		Species	Class	Order	Family	ID
subgroup_1	subgroup_2	outgroups				
		*Vulcanolepas fijiensis* (Chan, Ju & Kim, 2019)	Thecostraca	Scalpellomorpha	Neolepadidae	MN061491.1
		*Austinogebia edulis* (Ngoc-Ho & Chan, 1992)	Malacostraca	Decapoda	Upogebiidae	JN897376.1
		*Typhlatya galapagensis* (Monod & Cals, 1970)			Atyidae	KX844711.1
		*Munidopsis ennell* (Benedict, 1902)			Munidopsidae	MH717896.1
		*Longpotamon parvum* (Dai & Song in Dai, Song, Li, Chen, Wang & Hu, 1985)			Potamidae	MN737134.1
		*Metaplax longipes* (Stimpson, 1858)			Varunidae	NC_040976.1
		*Scyllarides squammosus* (H. Milne Edwards, 1837)			Scyllaridae	NC_044425.1
		*Artemia tibetiana* (Abatzopoulos, Zhang & Sorgeloos, 1998)	Branchiopoda	Anostraca	Artemiidae	JQ975177.1
		*Daphnia pulex* (Leydig, 1860)		Anomopoda	Daphniidae	NC_000844.1
		Compete				
		*Lepeophtheirus salmonis* (Krøyer, 1837)	Copepoda	Siphonostomatoida	Caligidae	NC_056769.1
		*Pennell sp.* (Oken, 1815)			Pennellidae	ON161759.1
		*Tigriopus japonicus* (Mori, 1938)		Harpacticoida	Harpacticidae	AB060648.1
		*Tigriopus kingsejongensis* (Park, S. Lee, Cho, Yoon, Y. Lee & W. Lee, 2014)				MK598762.1
		*Tigriopus californicus* (Baker, 1912)				NC_008831.2
		*Amphiascoides atopus* (Lotufo & Fleeger, 1995)			Miraciidae	NC_023783.1
		*Paracyclopina nana* (Smirnov, 1935)		Cyclopoida	Cyclopettidae	EU877959.1
		*Lamproglena chinensis* (Yü, 1937)			Lernaeidae	OQ411234
		*Lamproglena orientalis* (Markevich, 1936)				OQ411235
		*Lernaea cyprinacea* (Linnaeus, 1758)				NC_025239.1
		*Sinergasilus polycolpus* (Markevich, 1940)			Ergasilidae	NC_028085.1
		*Sinergasilus undulatus* (Markevich, 1940)				NC_054173.1
		*Sinergasilus major* (Markevich, 1940)				OQ160840
		*Calanus hyperboreus* (Krøyer, 1838)		Calanoida	Calanidae	NC_019627.1
		*Eurytemora affinis* (Poppe, 1880)			Temoridae	NC_046694.1
		*Phyllodiaptomus tunguidus* (Shen & Tai, 1964)			Diaptomidae	NC_046743.1
		*Calanus simillimus* (Giesbrecht, 1902)			Calanidae	NC_063666.1
		*Labidocera rotunda* (Mori, 1929)			Pontellidae	NC_064109.1
		partial				
		*Caligus rogercresseyi* (Boxshall & Bravo, 2000)	Copepoda	Siphonostomatoida	Caligidae	HQ157565.1
		*Caligus clemensi* (Parker & Margolis, 1964)				HQ157566.1
		*Schizopera knabeni* (Lang, 1965)		Harpacticoida	Miraciidae	KF667527.1
		*Calanus sinicus* (Brodsky, 1962)		Calanoida	Calanidae	GU355641.1
		*Calanus glacialis* (Jaschnov, 1955)				MF422146.1
		*Calanus finmarchicus* (Gunnerus, 1770)				MG001887.1
		*Lovenula raynerae* (Suárez-Morales, Wasserman & Dalu, 2015)			Diaptomidae	MH710604.1

**Table 2 genes-14-01496-t002:** K-scale factor and concatenated length for individual mitochondrial genes (tRNAs as the data set).

Comparison_Tree	K-Score	Concatenate Length
*rrnS*	2.717	1096
*rrnL*	2.686	1704
tRNAs	1.441	1092
*nad4L*	1.231	406
*nad3*	1.115	391
*nad6*	0.951	635
*cox3*	0.913	825
*nad4*	0.893	1438
*nad5*	0.799	1890
*cox2*	0.775	726
*cytb*	0.768	1191
*atp6*	0.738	720
*cox1*	0.712	1589
*nad1*	0.692	993
*nad2*	0.624	1123

## Data Availability

The genome sequence data that support the findings of this study are openly available in GenBank of NCBI at https://www.ncbi.nlm.nih.gov/ (accessed on 1 June 2022), reference number OQ160840.

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
