# Peer review of "Evolutionary Rates, Divergence Rates, and Performance of Individual Mitochondrial Genes Based on Phylogenetic Analysis of Copepoda"

_genes, 2023, doi:10.3390/genes14071496_

Round 1

Reviewer 1 Report

The authors present a well-elaborated work on the comparative analysis of the copepod mitochondrial gene. However the text needs improvement as the work initially seems to be just about Sinergasilus major and its impact on phylogeny. And throughout the discussion of the manuscript I feel that it is ignored. The work could have been done without his inclusion and would not change the outcome of copepod phylogeny and mitochondrial genome analysis. I think it would be good to reflect a little on the importance given to this species. It may have been included in the analysis, without having so much focus on it. The title presented should also be rethought. Will it focus on the phylogeny of the group or on the species Sinergasilus major?

The data obtained is very good, but there seems to be a dispersion, I advise the authors to compare the Copepoda general genome and leave the species S. major as just one more taxon added in the general analysis of the manuscript.

Some comments are in the pdf file.

Author Response

Thank you for your distinguished suggestions for our manuscript, which have greatly helped both revising this article and providing novel insights for our future research. We have made significant adjustments to the deconstruction of the manuscript. As most of our team is preparing for the post-graduate entrance examination in CN, the speed of revision is relatively slow, sorry for any impact that may cause. 
Please see the attachment for answers and tracking of the changes. 
Regarding the intention of the article, indeed adding a single species is insufficient to warrant a phylogenetic analysis of such a large lineage as Copepoda, but we want to focus on finding the differences and advantages in each mitochondrial gene. Phylogeny of Copepoda is just a comparison and summary of previous research.
This is not the final version as the suggestions from other reviewers have not been resolved. If you have further suggestions, please do not hesitate to contact us.

Reviewer 2 Report

In this work, the authors present the newly sequenced mitochondrial genome of a copepod Sinergasilus major. Their findings are sound and undoubtedly have scientific value. I especially liked the fact that the authors did not only analyze the new genome but did quite an extensive work comparing it with all the previously published mitochondrial genomes of this group.

However, I have some questions feel that the presentation can be further improved.

=== Major issues

Point 1: I couldn’t find information that the obtained raw sequence data and the assembled mitogenome were submitted to NCBI or alternative database. This is extremely important for the transparency, reproducibility and further citability of the study.

Point 2: in the end of the “Results and Discussion” section, the authors reconstruct the trees for each of the genes, note which of them match with the topology of the reference tree, but do not show the actual trees. It would be nice to show the trees in the manuscript. I would suggest adding the trees for cox1 and nad2 (in the same scale) as a figure in the main manuscript, and show the other genes in the supplement, but it’s totally up to the authors.

=== Text-related issues

Point 3:
L38: I don’t really agree with the combination of the text “use sequences from multiple loci are advantageous for study” and the reference the authors provide for this thesis, even though I agree with the thesis and do not have anything against the reference. As I understand, the source places a greater emphasis on mitochondrial+nuclear marker combinations. It says: “multiple sequences including some from the nuclear genome are needed to avoid artifacts from incomplete lineage sorting, hybridization, sequence paralogy and other problems.”
For example, full mitochondrial genomes do not solve the problem of ILS and hybridization.
So, it would be nice if the authors elaborate a little bit more on the advantages of full mitochondrial genomes compared to short marker fragments (without mixing this up with mito+nuclear markers) in addition to the advantages of mito+nuclear combinations to mito-only, especially taking into account that they later use 18S as a barcode.

Point 4:
L52: It would be great to more precisely name the host group, if it is possible. Is it only fish, mostly fish, or many other groups too?

Point 5:
around L95: How many samples were actually used in this work? Is 18S a well-established barcode for Sinergasilus species, and if yes, how do the 18S haplogroups correspond to cox1 haplogroups?

Point 6:
L139: what is Normal model in MAFFT? Does it mean the default settings?

Point 7:
L140: 36 genes (13 PCGs + 2 rRNA + 22 tRNA) should be 37

Point 8:
L145 Morderfinder is probably Modelfinder

Point 9:
L182: “celeste shading”? I see black, and celeste should be sky blue.

The manuscript needs to be proofread before it can be published. There are multiple tiny issues, which sometimes interfere with the reading.

Here are some minor spelling suggestions (I definitely haven’t spotted everything!)
L34-35 “The Metazoan mitogenomes” : metazoan (no need for a capital letter)
L38: “use sequences from multiple loci are advantageous for study” => “usage of sequences from multiple loci is advantageous for study” ? In addition, study of what?
L66: add some linking phrase to the beginning at the sentence like “At the same time”
L101: “The GenBank acces-
sion number for the isolate 18S gene determined in this work is OP076956.1.” “The GenBank accession number for the 18S gene fragment of the isolated used in this work for mitochondrial DNA sequencing is OP076956.1.”
L108
“with other Copepoda PCGs sequences from the Genbank database (available at https://www.ncbi.nlm.nih.gov/genbank/).” : would be nice to cite Table 1 right here
L124-125: “Nucleotide diversity
(Pi) of each PCGs and rRNA gene with a sliding window of 200 bp at a step size of 20 bp.”: a verb is missing, probably “was identified / estimated”
L161-162  “due to the complete mitogenome of Paracyclopina nana has lacked atp8 gene this
part atp8 gene will not be discussed.”
L165 “Result and discussion” => “Results and discussion”
L171 “J-strand” should probably be “H-strand”
L172 “the remain 26 genes,” => “the remaining 26 genes”
L187 ” were shortly (0 to 29 bp) in length” => “were short”
L194-195 Leucine and other amino acids do not require capital letters
L254-255 “has the slowest-evolving and least variable gene. ”  => “was”
L322-324 “Although sequencing a complete or near-complete mitogenome is neither difficult nor expensive along with advances in next generation sequencing platforms and bioinformatics tools” : the sentence is unfinished.

Author Response

Thank you for your distinguished suggestions for my manuscript. We have made significant adjustments to the structure of the manuscript. As most of our team is preparing for the post-graduate entrance examination in CN, the speed of revision is relatively slow, sorry for any impact that may cause.
Please see the attachment for answers and tracking the changes in the latest (only one file can be uploaded). This is not the final version as the suggestions from other reviewers have not been resolved. If you have further suggestions, please do not hesitate to contact us.

Reviewer 3 Report

Review of the paper: Comparative analysis of mitochondrial genomes of Copepoda and phylogenetic implications, submitted by Jun-zong He, Zhi-hao Zhou, Yan Huang, Jin-mei Feng, Wen-xiang Li, Gui-tang Wang and Cong-jie Hua

Major comments:

In the Discussion section there are several details using wrong labelling of some taxa (orders, families) along with wrong spelling of some names. Must be corrected.

The whole text must be corrected by native English speaker, particularly the last chapter.

There is a lot of nice statistical analyses but several important information are missing:

A)   Were all specimens included into the study from the same host fish? Was any time difference/seasons in sampling parasites?

B)    How many specimens were analysed?

C)    What was intraspecific variation with the genom of analysed specimens? – see lines 203-209.

D)    Are there any evidence for differences in “strong and relaxed purifying selection” between freshwater and marine species? However, most of species from Table 1 have marine origin.

Minor comments:

See attached PDF file.

Final conclusion: major revision.

I am not native English speaker, but I find language quite good, but some details (see comments in the PDF file). However, would be fine to be checked by native English speaker. 

Author Response

Thank you for your distinguished suggestions for my manuscript again, I have already contacted the editor. As most of our team is preparing for the post-graduate entrance examination in CN, the speed of revision is relatively slow, sorry for any impact that may cause. 
The relevant answers are as follows:
A and B: We are only using one specimen in the mitogenome sequencing, all samples are taken at the same time.
C: Since we only analyzed one specimen, we could not test for intraspecific variation
D: Species in freshwater and marine were not discussed in our study, but this may be a novel direction. The fact that most species in Table 1 have marine origin may be caused by the fact that mitochondrial genomes have been sequenced for more marine than freshwater species.

And other questions and revises you can track in Word. Regarding the supplementary documents, we initially displayed that the upload was successful. 
This is not the final version as the suggestions from other reviewers have not been resolved. If you have further suggestions, please do not hesitate to contact us.

Round 2

Reviewer 2 Report

The authors did a great job improving the article. Thank you!

It is very good that the data is now available at NCBI. However, I have a follow-up question. When I analyzed the initially submitted paper, I didn’t fully realize that the mitogenome was sequenced with the Sanger method (that’s why I had asked for both raw and assembled data). Is it possible to add the information on the primer set used for amplification and sequencing for full reproducibility?

English language looks fine to me now.

Author Response

Thank you for your suggestion. We add the information in new Table S1